# Reversal of carbonate-silicate cation exchange in cold slabs in Earth's lower mantle

Mingda Lv [1✉], Susannah M. Dorfman [1✉], James Badro [2], Stephan Borensztajn[2], Eran Greenberg [3,4] & Vitali B. Prakapenka [3]

The stable forms of carbon in Earth's deep interior control storage and fluxes of carbon through the planet over geologic time, impacting the surface climate as well as carrying records of geologic processes in the form of diamond inclusions. However, current estimates of the distribution of carbon in Earth's mantle are uncertain, due in part to limited understanding of the fate of carbonates through subduction, the main mechanism that transports carbon from Earth's surface to its interior. Oxidized carbon carried by subduction has been found to reside in $MgCO_3$ throughout much of the mantle. Experiments in this study demonstrate that at deep mantle conditions $MgCO_3$ reacts with silicates to form $CaCO_3$. In combination with previous work indicating that $CaCO_3$ is more stable than $MgCO_3$ under reducing conditions of Earth's lowermost mantle, these observations allow us to predict that the signature of surface carbon reaching Earth's lowermost mantle may include $CaCO_3$.

[1] Department of Earth and Environmental Sciences, Michigan State University, East Lansing, MI, USA. [2] Université de Paris, Institut de physique du globe de Paris, CNRS, Paris, France. [3] Center for Advanced Radiation Sources, University of Chicago, Chicago, IL, USA. [4] Present address: Applied Physics Department, Soreq Nuclear Research Center (NRC), Yavne 81800, Israel. ✉email: lyumingd@msu.edu; dorfman3@msu.edu

Carbon is not the only key to life and Earth's habitability but also traces and modifies geological processes of sub-duction, partial melting, degassing, and metasomatism, providing valuable insights into Earth's evolution[1]. Over the history of the planet, carbon transport between surface and deep reservoirs has impacted the atmospheric, oceanic, and crustal $CO_2$ budgets in tandem with the composition and redox state of the Earth's mantle[2,3]. Carbon is transported from Earth's surface to its interior mainly as carbonate minerals in subduction zones and is returned in carbon-bearing gas/fluid through volcanic degassing[2,3]. These processes leave signatures in the mantle, including depletion of incompatible elements[4,5], diamond for-mation (and inclusions)[6,7], and isotopic abundances[8,9]. Carbon flux via subduction to the deep mantle remains uncertain, with estimated magnitudes ranging from 0.0001 to 52 megatons/year[3,10]. The wide range of these estimates is due in part to limited understanding of the physical and chemical responses of carbonates to mantle pressures, temperatures, and compositional environments.

The dominant carbonates carried into the mantle by subducting slabs, dolomite $CaMg(CO_3)_2$, magnesite $MgCO_3$, and calcite $CaCO_3$[11], undergo changes in crystal structure or state and che-mical reactions at depth. Carbonates are likely to be retained as solid minerals in subducting ocean crust until/unless the solidus of carbonated peridotite[12,13] or eclogite[14,15] intersects with mantle geotherms, initiating melting. These slab-derived carbonatite melts will segregate to the overlying mantle due to low viscosity and density[16], or be reduced to diamonds at depths greater than ~250 km via redox freezing[7,15,17]. However, carbonates are present in the mantle transition zone and possibly lower-mantle depths in some regions, based on direct evidence provided by carbonate minerals found in deep-sourced diamond inclusions[18,19]. Addi-tional evidence from thermodynamic modeling of devolatilization of carbonate-bearing subducting slab[20,21] and melting experi-ments on carbonates in the $MgCO_3$–$CaCO_3$ system up to 80 GPa[22] supports the preservation of solid carbonates along low-temperature geotherms in subducting slabs in the lower mantle. However, the temperature is not the only control on the fate of subducted carbonates: carbonates may also interact chemically with the major phases of the ambient mantle or basalt-rich sub-ducted crust. In these compositions in the lower mantle, the sili-cates potentially reacting with carbonates are bridgmanite (bdg), post-perovskite (pPv), and Ca-perovskite (Ca-Pv).

The presence of the end-member carbonates, $MgCO_3$ and $CaCO_3$ (note that $(Mg,Ca)(CO_3)_2$ dolomite breaks down to these end-members above 5 GPa and 1200 K[23]), together with lower-mantle silicates depends on the thermodynamics and kinetics of the carbonate–silicate exchange reaction:

$$CaCO_3 + MgSiO_3 \rightarrow MgCO_3 + CaSiO_3 \qquad (1)$$

Previous experiments[24,25] indicate that $CaCO_3$ reacts with silicates to form $MgCO_3$ via the forward reaction up to 80 GPa and 2300 K, i.e., at least to the mid-lower mantle. Theoretical studies further predict that $MgCO_3 + CaSiO_3$ are enthalpically favored over $CaCO_3 + MgSiO_3$ throughout the lower mantle pressure and temperature regime[26–30]. However, although many studies have addressed the stability of individual carbonates up to higher pressures[31–33], no experiments examined the carbonate–silicate cation exchange reaction up to core–mantle boundary conditions.

In this work, to assess the stability of $MgCO_3$ and $CaCO_3$ coexisting with lower-mantle silicates, we conduct a series of experiments on the carbonate–silicate reaction along the lower-mantle geotherm. Thin disks of carbonates and silicates were loaded together in laser-heated diamond-anvil cells (LHDAC, Supplementary Table 1, see "Methods" for details). Laser heating

at 1600–2800 K and 33–137 GPa was applied for 10–400 min. Run products were examined by in situ synchrotron X-ray dif-fraction (XRD) and ex situ energy-dispersive X-ray spectroscopy (EDX) analysis with a scanning transmission electron microscope (STEM, see "Methods" for details).

## Results

**Calcium carbonate reaction to form magnesium carbonate.** Experiments assessed thermodynamic stability by using as reac-tants either $(Mg,Ca)CO_3 + (Mg,Fe)SiO_3$ (reactants for the for-ward reaction, hereafter referred to CaC-to-MgC) and $(Mg,Fe)CO_3 + CaSiO_3$ (reactants for the reverse reaction, hereafter referred to MgC-to-CaC). For reaction CaC-to-MgC, the criterion for determining whether the reaction takes place is the presence of newly synthesized $CaSiO_3$-perovskite in the run product. For reaction MgC-to-CaC, newly synthesized $MgSiO_3$ and $CaCO_3$ indicate the reaction is favorable. The silicate reaction products are easier to observe through diffraction than carbonates due to higher diffraction intensity.

Experiments with CaC-to-MgC reactants indicate the forward reaction takes place in runs conducted below 83 GPa (runs #1–4), as determined via both EDX and XRD. For example, ex situ electron microscopic analysis of the sample recovered from 33 GPa and 1650 K (run #1) (Fig. 1a–c) reveals a ~1-μm-thick layer of $CaSiO_3$ between the silicate layer and the carbonate layer, coexisting with $SiO_2$, FeO, $MgSiO_3$, and $MgCO_3$. These observations are consistent with in situ XRD patterns of run products after heating (Supplementary Figs. 3b and 5a), which exhibit several new sharp peaks compared to the pattern before heating (Supplementary Fig. 3a). The diffraction pattern of run products is consistent with the presence of Ca-Pv, magnesite, bdg, wüstite, stishovite, and monoclinic dolomite III (previously observed at pressure above 36 GPa[34]). Ca-Pv can be observed in the run products of CaC-to-MgC up to 83 GPa (Supplementary Figs. 3c and 4a, b), in agreement with previous experimental observations[24,25].

At higher pressures from 91 to 137 GPa, however, we observe no evidence of carbonate–silicate exchange reaction in experiments with CaC-to-MgC reactants. Ca-Pv is not identified in the run products (runs #5-7) through either in situ (Supplementary Figs. 3d, e, 4c, d, and 5b) or ex situ analysis. New, sharp peaks from bdg and pPv can be observed in situ in XRD patterns (Supplementary Fig. 3d, e), indicating the sample was sufficiently heated to transform starting materials to high-pressure silicate structures, but no carbonate–silicate exchange reaction occurs. Two hypotheses can explain these observations: (1) in contrast to theoretical predictions that the reversal of the carbonate-exchange reaction takes place at higher pressures and lower temperatures[26–30], $CaCO_3 + MgSiO_3$ become more favorable than $MgCO_3 + CaSiO_3$ from 91 to 137 GPa and 2100 to 2800 K; (2) the reaction CaC-to-MgC is hindered by reaction kinetics, and metastable starting materials are observed.

**Magnesium carbonate reaction to form calcium carbonate.** In order to resolve the thermodynamically stable phase assemblage, three separate sets of experiments on the backward reaction (MgC-to-CaC, runs #8–11) were conducted at 35–133 GPa and 1800–2000 K. Elemental mapping of the run products of experi-ments at 88 GPa (run #10, Fig. 1d–f) and 133 GPa (run #11, Fig. 1g–i) indicates that $MgSiO_3$ layers formed along with the carbonate–silicate interface, and newly formed $CaCO_3$ can be observed as well. At 35 GPa, neither EDX nor XRD shows $MgSiO_3$ formed from MgC-to-CaC reactants (run #8, Supple-mentary Fig. 9). Observations of the reversal of the reaction confirm that $MgCO_3$ is unstable and reacts with $CaSiO_3$ produ-cing $CaCO_3$ and $MgSiO_3$ at pressures higher than 88 GPa along a lower-mantle geotherm.

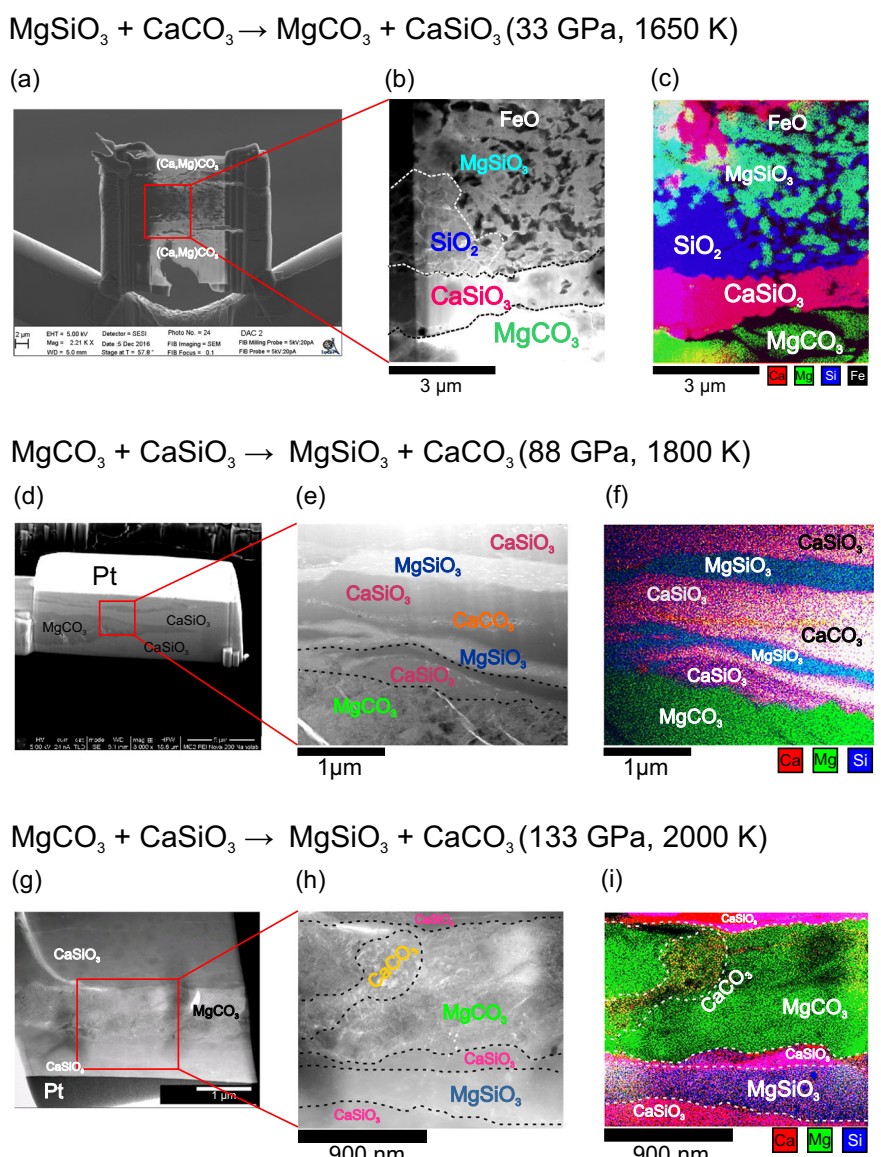

**Fig. 1 Electron microscopic characterizations of recovered samples.** Images of selected recovered sample cross-sections obtained using backscattered scanning electron microscopy (**a**, **d**, **g**), scanning transmitted electron microscopy (**b**, **e**, **h**), and energy-dispersive X-ray mapping (**c**, **f**, **i**) of the cross-section show the silicate layer sandwiched by two carbonate layers, with the reaction region along the contacting interface. **a**–**c** Ex situ analysis of sample quenched from 33 GPa and 1650 K heated for 15 min (run #1) demonstrates reaction CaC-to-MgC: $CaSiO_3$ is not present in starting materials but is indicated in EDX map by colocation of Ca and Si, shown in magenta. **d**–**f** Ex situ analysis of sample quenched from 88 GPa and 1800 K heated for 150 min (run #9) demonstrates reaction MgC-to-CaC: $MgSiO_3$ is not present in starting materials but is indicated in EDX map by colocation of Mg and Si, shown in blue-green. $CaCO_3$ also appears as a red (Ca, but no Si) ribbon within $CaSiO_3$ starting material. **g**–**i** Ex situ analysis of sample quenched from 133 GPa and 2000 K heated for 400 min (run #10) demonstrates reaction MgC-to-CaC: $MgSiO_3$ appears as Ca-depleted, Si-rich region (blue or blue-green) adjacent to $CaSiO_3$ starting material (magenta).

Our results agree with previous experimental constraints (Fig. 2) below 80 GPa showing: dolomite is unstable relative to $CaCO_3$ and $MgCO_3$ at lower-mantle conditions[23–25,35]; neither CaO nor MgO are observed in run products, indicating no decomposition of $CaCO_3$ and $MgCO_3$ into oxides plus $CO_2$[26,27,30]; $MgCO_3$ is more favorable in the lower mantle than $CaCO_3$ up to ~80 GPa due to the CaC-to-MgC reaction[24,25]. Since similar previous studies were limited to pressures below 80 GPa, they did not observe the reversal reaction (MgC-to-CaC). Combining our new results with previous results[24,25] and theoretical predictions indicating a positive Clapeyron slope for this reaction[28–30], we suggest a reaction boundary above 80 GPa with a positive slope (black dashed line in Fig. 2). We note that the experimental data allow for significant uncertainty in this

boundary, but are inconsistent with theoretical predictions[28–30] (yellow region, Fig. 2). This discrepancy may have been produced by theoretical approximations at higher temperatures. If density functional perturbation theory and quasi-harmonic approximation have misestimated the volumes of the carbonate phases expected to be stable at ~80 GPa and higher pressures, this could lead to the systematic overestimation of Gibbs free energy of $CaCO_3 + MgSiO_3$ relative to $MgCO_3 + CaSiO_3$ at higher temperatures.

## Discussion
The pressure/temperature conditions of the reversal reaction as constrained by these experiments are similar to those of

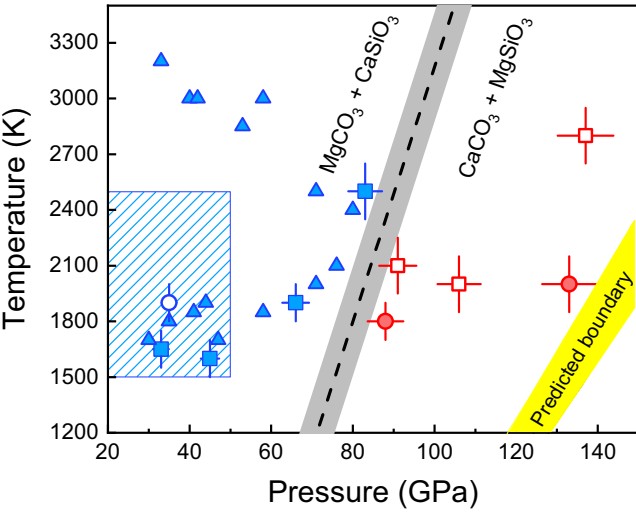

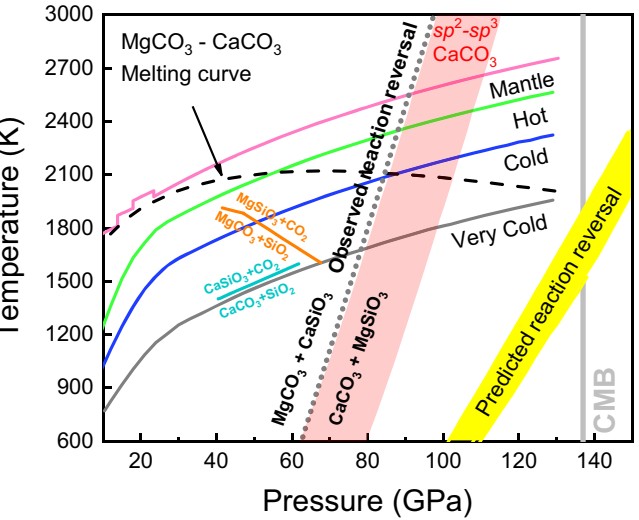

**Fig. 2 Phase diagram for the relative stability of the MgCO₃ + CaSiO₃ assemblage versus CaCO₃ + MgSiO₃.** The boundary sketched as a black dashed line with gray shadow as uncertainty inferred is based on experimental observations of carbonate–silicate exchange reactions CaC-to-MgC and MgC-to-CaC. Squares represent observations from this work starting with (Ca,Mg)CO₃ and (Mg,Fe)SiO₃, looking for newly synthesized CaSiO₃ to indicate the CaC-to-MgC reaction takes place. Circle symbols represent observations from this work of experiments starting with (Mg,Fe) CO₃ + CaSiO₃, looking for identification of newly synthesized MgSiO₃ to indicate the MgC-to-CaC reaction takes place. Open symbols indicate nonreaction and filled for confirmed reaction, and blue and red colors correspond to the inferred stable phase assemblage based on reaction products. Triangles indicate the P–T conditions for CaC-to-MgC taking place reported by Seto et al.[25], and blue-shaded region indicates approximate conditions of four experiments conducted by Biellmann et al.[24] using indirect methods for pressure and temperature calibration, which all produced the run products MgCO₃ + CaSiO₃. The error bars indicate uncertainties of pressure and temperature measurements (see "Methods" for details). The boundaries proposed by previous theoretical predictions are illustrated by yellow-shaded region[28–30].

**Fig. 3 Pressure–temperature diagram of reactions between carbonate, silicates, and silica in the subducted oceanic crust to the lower mantle.** The gray dotted line indicates the reversal boundary of the carbonate–silicate exchange reaction proposed by this study, whereas previous theoretical predictions are illustrated by yellow-shaded region[28–30]. The cyan and orange lines indicate the decarbonation reactions of CaCO₃ + SiO₂[40] and MgCO₃ + SiO₂[41], respectively. The black dashed line shows the melting curve of MgCO₃–CaCO₃ system constrained by Thomson et al.[22]. Four typical mantle geotherms are modified from Maeda et al.[36]. The red-shaded region indicates the transition boundary of CaCO₃ from sp² to sp³ structure predicted by density functional theory computations[29,38].

polymorphic phase transitions associated with $sp^2$–$sp^3$ bonding changes in both MgCO₃ and CaCO₃, which suggests these transitions are related to the stabilization of a CaCO₃ + MgSiO₃ assemblage. The transition from $sp^2$- to $sp^3$ bonds in MgCO₃ has been identified at ~80 GPa with the stabilization of the $C2/m$ structure[32,33,36], and the resulting densification of MgCO₃ supports the forward reaction to MgCO₃ + CaSiO₃. The transition in CaCO₃ from $sp^2$- to $sp^3$ bonds in the $P2_1/c$-h structure was experimentally observed at ~105 GPa and 2000 K[37]. Computational studies predicted this boundary at ~70[29] and ~100 GPa[38] at mantle-relevant temperatures (red-shaded region in Fig. 3). While an earlier study that did not include the $sp^3$ CaCO₃-$P2_1/c$-h structure predicted a crossover in silicate–carbonate-exchange reaction at 135 GPa and 0 K[26], a later study that predicted the $sp^3$ CaCO₃-$P2_1/c$-h structure found a silicate–carbonate reaction reversal at 84 GPa and 0 K[30]. This would correspond to $sp^2$–$sp^3$ crossover and stabilization of CaCO₃ + MgSiO₃ in the mid-lower mantle.

Whether a crossover in the carbonate–silicate exchange reaction takes place in the deep Earth depends on whether carbonates are preserved in Earth's lower mantle to at least 1800-km depth. Previous studies have identified barriers to carbon subduction and stability in the lower mantle, particularly melting[15,39] and reduction[40–42]. If carried in cold subducting slabs, MgCO₃ and CaCO₃ may avoid melting as their melting temperatures[22] are higher than some predicted cold slab geotherms[36]. Any solid

carbonate in the mantle will be in contact and may equilibrate with silicates in all mantle environments and with free silica in basalt-rich compositions. MgCO₃ and CaCO₃ have been observed in experiments[40–42] to undergo decarbonation reactions with free silica over a pressure range of ~40 to 60 GPa. However, the Clapeyron slope of CaCO₃ + SiO₂ → CaSiO₃ + CO₂ is positive and takes place at pressure/temperature conditions warmer than the coolest slab geotherms[40]. Observations that MgCO₃ is less thermally stable than CaCO₃ support the survival of CaCO₃ rather than MgCO₃ along a cold subducted slab geotherm to the lowermost mantle[36,41] (Fig. 3). In this study, we report a reversal in the Mg–Ca silicate–carbonate cation exchange reaction at ~90 GPa, making MgCO₃ + CaSiO₃ favorable in the upper part of the lower mantle, while CaCO₃ + MgSiO₃ is preferred in the lower part of the lower mantle (Fig. 3). However, the question of whether any carbonate persists to these depths in the coldest subducting slabs remains unresolved. If MgCO₃ remains present in cold slabs, and the reaction CaC-to-MgC proceeds throughout most of the mantle eliminating CaCO₃, the reversal MgC-to-CaC reaction may transform MgCO₃ back to CaCO₃ in the lowermost mantle (Fig. 4). CaCO₃ could thus be found in the lowermost mantle coexisting with silicates and reduced iron.

The reduced nature of the Earth's mantle, with oxygen fugacity inferred to be near the iron–wüstite buffer in the transition zone and greater depths[43], stabilizes diamond or Fe carbide as long-term hosts of carbon, owing to their chemical refractoriness and dynamic immobility[44]. Similarly, our experimental observations support CaCO₃ as a refractory, stable host for oxidized carbon in the middle to the lowermost mantle, in particular, the high-pressure polymorph of CaCO₃ (CaCO₃-$P2_1/c$-h) with tetrahedral bonds[37]. Experimental observations also suggest that CaCO₃ is more resistant to redox breakdown reaction with iron under reduced conditions than MgCO₃[35]. In addition, due to the cation

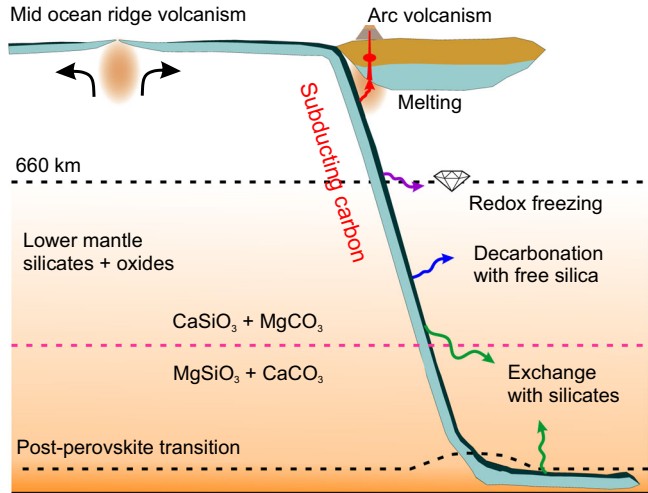

**Fig. 4 Schematic illustration of the fate of carbonates in the oceanic crust (dark blue) subducted to the lower mantle.** Through subduction, the carbonates may undergo melting (red arrow), redox freezing with metallic iron (purple arrow), decarbonation reaction with free silica (blue arrow), and exchange reaction with lower-mantle silicates (green arrow). Based on the observation of reversal of the carbonate–silicate cation exchange reaction at conditions relevant to cold subducted slabs at mid-lower-mantle depths, $CaCO_3$ is the potential stable phase that hosts oxidized carbon in the lowermost mantle.

exchange between carbonate and silicate, the relative stability of $MgCO_3$ or $CaCO_3$ will change in the lowermost mantle, and depending on conditions one of these phases may buffer the redox state of the mantle through an influx of oxidized carbon in the form of solid carbonate[45].

The Mg–Ca silicate–carbonate-exchange reactions along with subduction pressure–temperature (P–T) conditions may impact observable signatures of Mg and Ca isotopes in mantle silicates under certain special conditions, or in carbonate inclusions in diamonds. Subducting carbonates carry low-$\delta^{44/40}$Ca and low-$\delta^{26}$Mg signatures relative to the heavier mantle ratios, but although previous studies have observed heterogeneity in the Ca and Mg isotope signatures in basalts and mantle peridotites, these studies determined that lighter ratios cannot be simply interpreted as evidence of recycled marine carbonates[46,47]. The Mg–Ca silicate–carbonate-exchange reactions along with subduction P–T conditions may contribute to these variable Mg and Ca isotopic compositions. The reaction CaC-to-MgC in the transition zone and upper part of the lower mantle would transfer light Ca isotopes from subducted $CaCO_3$ to $CaSiO_3$ (Ca-Pv) (Supplementary Note 1 and Supplementary Fig. 12). Isotopically light Ca-Pv can then be trapped in diamond inclusions and return to the surface[48], while the Ca isotopic signature of upwelling rocks would remain variable, as it undergoes continuous fractionation within peridotitic mantle lithologies[46,49–51]. The modification of carbonate–silicate phase equilibria observed in this study provides a new process that could alter Mg and Ca isotopic composition in such lithologies (Supplementary Note 1 and Supplementary Fig. 12). While the isotope signature of $MgSiO_3$ produced by reaction MgC-to-CaC would not be observable due to the small masses involved relative to the vast lower-mantle reservoir of $MgSiO_3$, any $CaCO_3$ produced in the deep lower mantle by this reaction would carry a heavier deep mantle $\delta^{44/40}$Ca signature that would distinguish it from surface-derived carbonate. If preserved in diamond inclusions and returned to the surface, heavy $CaCO_3$ could be used to trace the presence of oxidized carbon in the lowermost mantle. The potential of $CaCO_3$

to be a signature of an ultradeep carbon cycle reaching the core–mantle-boundary region may help to reveal other mysteries of the deep mantle, such as heat budget related to radioactive elements stored in Ca-bearing silicates[52], and compositions of heterogeneities that may record Earth's early history[48,53].

## Methods

**Starting materials**. To investigate phase equilibria in the carbonate–silicate system in Earth's lower mantle and control for effects of reaction kinetics, both CaC-to-MgC and MgC-to-CaC experiments were carried out in symmetric diamond-anvil cells (DAC) with flat-top double-sided laser heating[54]. For CaC-to-MgC, natural dolomite with homogeneous composition of $(Mg_{0.38}Ca_{0.59}Fe_{0.03})CO_3$ was used as a carbonate reactant, the composition, and structure of which has been characterized by X-ray fluorescence spectroscopy and X-ray diffraction, respectively[35]. Fe-bearing enstatite synthesized at École Polytechnique Fédérale de Lausanne with a composition of $(Mg_{0.5}Fe_{0.5})SiO_3$ was used as a silicate reactant[55]. For MgC-to-CaC, natural ferromagnesite (sample from Princeton University) was used as a carbonate reactant, with composition determined to be $(Mg_{0.87}Fe_{0.13})CO_3$ by wavelength dispersive X-ray spectroscopy in a Cameca SX100 Electron Probe Microanalyzer at the University of Michigan. Pure calcium silicate ($CaSiO_3$, Alfa Aesar) was used as a silicate reactant. The chief advantages to the abovementioned starting compositions are that recognition of a carbonate–silicate exchange reaction only requires identification of the presence of newly synthesized silicates in quenched run products, i.e., Ca-perovskite (Ca-Pv) in CaC-to-MgC and bridgmanite (bdg) in MgC-to-CaC; and Fe-bearing enstatite and ferromagnesite can serve as laser absorber during the forward CaC-to-MgC and reversal MgC-to-CaC experiments, respectively.

**LHDAC experiments**. The dolomite, enstatite, and calcium silicate samples were separately ground under acetone in an agate mortar for ~2 h each to achieve homogenous, finely powdered samples with grain size typically less than ~2 μm. A single ferromagnesite crystal was double-side polished to ~10-micron thickness. All starting materials were dried in an oven at 120 °C overnight before loading, and the powder samples were subsequently pressed in a DAC to form thin foils approximately ~8–10-μm thick. The enstatite foils and ferromagnesite crystals were sandwiched between iron-free dolomite and calcium silicate, respectively, serving as thermal insulators in symmetric DACs for CaC-to-MgC and MgC-to-CaC (Supplementary Figs. 1 and 2). No other pressure standard or medium was loaded to prevent reactions with other components and contamination of the chemical system. The sample sandwiches were loaded in sample chambers with diameters approximately halves of the anvil culet sizes drilled into Re gaskets pre-indented to a thickness of ~30 μm, by using the laser drilling system at HPCAT (Sector 16) of the Advanced Photon Source (APS), Argonne National Laboratory (ANL)[56]. Diamond anvils with flat culets of 300 μm were used for experiments under 60 GPa, beveled culets of 150/300 μm for experiments under 100 GPa, and beveled culets of 75/300 μm for experiments up to 140 GPa.

Before laser heating, each sample was compressed to the target pressure at 300 K, and after heating each sample was quenched to ambient pressure at 300 K to limit and preserve reactions at target conditions. Pressures were determined from the Raman shift of the singlet peak of the diamond anvil at the culet surface[57], and post-heating pressures were typically within 3% of the pre-heating pressure. Thermal pressure during heating may be estimated to be ~10% GPa higher than the pre-heating pressure at the modest temperatures[58,59]. High-temperature conditions were achieved by using a double-sided ytterbium fiber laser heating system at beamline 13-ID-D (GeoSoilEnviroCars) of APS, ANL[54], with two 1.064 μm laser beams focused down to a flat-top spot with a diameter of 10–12 μm on both sides of the sample. Temperatures of the heated samples were determined by fitting the measured thermal radiation spectra using the Planck radiation function under the graybody approximation[54]. The temperature reported in Supplementary Table 1 is the temporal average of multiple temperature measurements over the heating duration. Temperature fluctuations over this timescale were less than the specified uncertainty, which is derived from a standard deviation of temperature measurements from both sides of the laser-heated sample (typically ± 100 K below 2000 K and ±150 K above 2000 K) (Supplementary Figs. 10 and 11). Experiments were held at temperatures between 1600 and 2800 K for ~30 min in CaC-to-MgC experiments and up to 400 min in MgC-to-CaC experiments.

**In situ XRD**. Phases synthesized at high P/T and achievement of chemical steady-state were determined by in situ angle-dispersive X-ray diffraction (XRD) measurements performed before, during, and after heating at beamline 13-ID-D (GeoSoilEnviroCars) of APS, ANL. The incident X-ray beam was focused down to less than $3 \times 4$ μm$^2$ with a monochromatic wavelength $\lambda = 0.3344$ Å. Diffracted X-rays were recorded using a MAR 165 detector or Pilatus 1 M CdTe pixel array detector. NIST standard $LaB_6$ was used to calibrate the detector distance, tilt angle, and rotation angle of the image plane relative to the incident X-ray beam. Exposure times were typically 30 s. The XRD patterns were integrated to produce $2\theta$ plots using the software DIOPTAS[60].

**Ex situ EDX**. After complete pressure release, each sample was recovered from the LHDAC, and then sectioned along the compression axis through the laser-heated spot and over the entire thickness of the DAC sample (~5–20 μm), using a focused ion beam (FIB) coupled with a field-emission scanning electron microscope (FE-SEM) at IPGP (Paris, France) or the Michigan Center for Materials Characterization at the University of Michigan (Ann Arbor, USA). A ~30-nm-thick Au layer was coated on each sample to reduce charging in the scanning electron microscope, and a 2-μm-thick Pt layer was deposited across the center of each heated spot to protect the sample from damage by the $Ga^+$ ion beam. Thin sections of each heated spot were extracted and polished to electron transparency (~100-nm thickness).

Textural and chemical characterization of recovered samples was performed with scanning transmission electron microscopy (STEM) and energy-dispersive X-ray spectroscopy (EDX) in a JEOL 2200FS field-emission TEM (Center for Advanced Microscopy, MSU), operated at 200 kV to image the sample in brightfield. EDX maps were scanned over $512 \times 384$ pixel areas with a pixel dwell time of 50 μs. Typical count rates were ~2000 counts per second. Chemical mapping rather than point measurement approach prevents migration of elements due to damage by the electron beam. Uncertainties in compositions were determined from standard deviations of EDX measurements obtained from selected regions within multiple grains.

## Data availability

Additional diffraction and spectroscopy data and metadata are available in the Supplementary Information and from the corresponding authors upon request.

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

## Acknowledgements

We thank Feng Huang and Shichun Huang for helpful discussions on isotopes. We thank A. Hunter (University of Michigan's Michigan Center for Materials Characterization), X. Fan (Center for Advanced Microscopy, Michigan State University), and O.K. Neill (Robert B. Mitchell Electron Microbeam Analysis Lab) for assistance with SEM-FIB, STEM, and EPMA, respectively. This work was supported by new faculty startup funding from Michigan State University, the Sloan Foundation's Deep Carbon Observatory Grant G-2017-9954, and National Science Foundation (NSF) grant EAR-1751664 to S.M.D. Parts of this work were supported by IPGP multi-disciplinary program PARI, and by Region Île-de-France SESAME Grant no. 12015908. GeoSoilEnviroCARS is supported by U.S. Department of Energy (DOE) award DE-FG02-94ER14466 and NSF grant EAR-1634415. The Advanced Photon Source, a DOE Office of Science User Facility, is operated by Argonne National Laboratory under contract DE-AC02-06CH11357.

## Author contributions

M.L.: conceptualization, investigation, formal analysis, writing—original draft, and visualization; S.M.D.: conceptualization, methodology, investigation, writing—review and editing, and supervision; J.B.: conceptualization, supervision, and writing—review and editing; S.B.: investigation; E.G.: resources and writing— review & editing; V.B.P.: resources, methodology, and writing—review and editing.

## Competing interests

The authors declare no competing interests.
