## [Peer Review File · Nature Communications]

REVIEWER COMMENTS

Reviewer #1 (Remarks to the Author):

The article by Lv et al reports new and exciting high pressure and temperature experiments on the reactivity of carbonates and silicates as well as on the stability of these phases. The experiments seem well done and the authors took great care to show that they are seeing what they say they are seeing in the experiments. While I am confident that the experiments are "correct" my main issue with the text is the implication section at the end of the manuscript. From lines 138 - 179, there are many high level comments that I do not think are implied from the data. For example, the sentence in line 155 about calcium carbonate being in the lowermost mantle coexisting with silicates and reduced iron follows a chain of sentences that begin or deal with assumptions that may or may not be true. Further, the following paragraphs are even more difficult to reconcile, the isotopic argument, for example does not really flow from the data. If the authors wanted to invoke isotope arguments then a simple mass balance calculation would have helped a lot. (As an aside - anomalies in isotopes refers to mass independent fractionation usually, though I think the authors are considering a stable isotope fractionation). Suggesting that Ca isotopes could be fractionated during upwelling at the CMB and then brought to the surface requires way more data or at least some evidence or calculation.

I do not think that my comments suggest that the paper is not correct or should not be published. I just think more care should be given to these very high level statements as the data in the paper do not support them.

Reviewer #2 (Remarks to the Author):

The authors describe novel chemistry of carbonates and silicates at mantle conditions with significant impact for our understanding of geological compositions. The experiments are excellent in their design and appear well-performed, but some improvements in transparency are required by the authors on this front.

There is a distinct lack of structural refinement in the article or its supplemental document. Instead, the authors show raw diffraction data alongside tick marks from simulated spectra. While it is appreciable that these systems are mixtures of low-symmetry phases, making structural refinement a challenge, it can also be argued that since these systems are mixtures of low-symmetry phases, a collection of tick marks placed alongside the raw spectrum is not sufficient to show agreement between the model and the observed data.

In addition, it should be common practice to show microscope images of loaded samples at pressure in diamond cell experiments. It is important that the integrity of the experiment can be subject to scrutiny by the referees and eventual readers, and the sample loading is central to the success and quality of any diamond cell experiment. These authors are at the top of the field of static high pressure, and I hope that they agree, and will provide these in their supplement and any future publications.

The work is scientifically sound from beginning to end, and has implications which are of broad enough interest for the readership of Nature Communications. The authors demonstrate suitable attention to the literature and provide proper discussion of their results in view of some of the most recent results in carbonate chemistry at these conditions. Overall, the work demonstrates the state-of-the-art for LH-DAC experiments, but not the cutting-edge. Based on the experimental findings, the work warrants publication in Nature Communications, I would only ask that – especially for a broad readership journal – the authors provide more detailed illustration and documentation of their experimental procedures for the non-specialist.

Some notes on that front:

- The aforementioned structural fitting of XRD data
- Images of sample loadings
- Representative Planck fitting for temperature derivation

Responses to reviews' comments

(Our replies are in blue and the revisions are in red)

Reviewer #1 (Remarks to the Author):

The article by Lv et al reports new and exciting high pressure and temperature experiments on the reactivity of carbonates and silicates as well as on the stability of these phases. The experiments seem well done and the authors took great care to show that they are seeing what they say they are seeing in the experiments.

Thanks for recognizing the importance and quality of our work.

While I am confident that the experiments are "correct" my main issue with the text is the implication section at the end of the manuscript. From lines 138 - 179, there are many high level comments that I do not think are implied from the data. For example, the sentence in line 155 about calcium carbonate being in the lowermost mantle coexisting with silicates and reduced iron follows a chain of sentences that begin or deal with assumptions that may or may not be true.

We agree with the reviewer that whether CaCO_3 truly resides in the lowermost mantle remains an open question, and that the logic for this possibility rests on several open questions. We have been careful throughout this paragraph to use soft language to emphasize that the presence of this phase in the real Earth remains only a possibility that depends on these assumptions. In addition, we provide references that assess the current state of knowledge and uncertainty about these assumptions. It is beyond the scope of this study to independently address melting and reactions with other mantle phases as barriers to carbon subduction, or new observations of the global carbon budget that could verify whether subduction currently carries carbonates to lower mantle depths. Our goal is to assess whether it is possible within the uncertainty of current literature that this reaction takes place in the real Earth (rather than to prove it must), and to provide guidance to future studies on resolving this open question. We revise the text to emphasize the purpose of this paragraph, starting line 148:

Whether this reaction takes place in the deep Earth depends on whether carbonates are preserved in Earth's lower mantle at least 1800 km depth. Previous studies have identified barriers to carbon subduction and stability in the lower mantle, particularly melting^{22,35} and reduction³⁸⁻⁴⁰. [detailed text] However, the question of whether carbonate persists in the coldest subducting slabs remains unresolved. If MgCO_3 remains present in cold slabs...

Further, the following paragraphs are even more difficult to reconcile, the isotopic argument, for example does not really flow from the data. If the authors wanted to invoke isotope arguments then a simple mass balance calculation would have helped a lot. (As an aside - anomalies in isotopes refers to mass independent fractionation usually, though I think the authors are considering a stable isotope fractionation). Suggesting that Ca isotopes could be fractionated during upwelling at the CMB and then brought to the surface requires way more data or at least some evidence or calculation.

We acknowledge that we do not contribute new isotopic data, but would like to be comprehensive in an assessment of **what observables could test for the presence of carbonates in the deep mantle**, and whether Ca and Mg isotopes can be used to trace the deep carbon cycle. We agree with the reviewer that the isotopic implications are not directly based on our data, but instead use related literature to make conjectures and suggest directions for future research. We thank the reviewer for the comments, which

demonstrated to us that we need to elaborate on a scenario based on our experiments that we suggest could impact observable isotope signatures, and the suggestion to add a mass balance calculation.

Our main goal is to direct future research to determine whether and how much carbonate may be present in the lower mantle. Although previous studies cited in the text suggested the possibility that isotopes would be fractionated during upwelling, we do not expect that the carbonate-silicate phase equilibria investigated in our study contribute to this fractionation, but rather that any fractionation that takes place during upwelling may modify signals produced at greater depth. In general and based on literature discussed in our Introduction, carbonates are most likely to be present within cold subducting plates, and would not be expected to be widespread throughout the warmer well-mixed part of the mantle. If carbonates in a slab do not completely melt or reduce to diamond/carbide, and reach the lowermost mantle, they still may not be present in upwelling mantle as carbonates, to undergo fractionation while upwelling. We assume that fractionation related to reactions with carbonates is most likely to take place within subducting slabs, or as carbonate in slabs reacts with surrounding mantle.

To test for carbonate in the lower mantle, we imagine two main possibilities:

- 1) Carbonate subducted to the lower mantle has a significant effect on Ca or Mg isotopes of the silicates in the mantle (this is the scenario for which a mass balance suggested by the reviewer would be helpful).

Light Ca and Mg isotopes are carried in carbonates in subducting slabs. In the upper mantle through shallow lower mantle, the light Ca isotopes brought by CaCO_3 would go into CaSiO_3 and be diluted by the mantle signature, while light Mg isotopes brought by MgCO_3 would remain in the carbonate. Based on our new experiments, in the deep lower mantle, MgCO_3 is unstable, so the light Mg isotopes brought by MgCO_3 may exchange into the surrounding ambient deep lower mantle and be diluted in the far greater reservoir of MgSiO_3 (amount of Mg in mantle is $\sim 10\times$ amount of Ca).

As suggested, we performed an isotopic mass balance calculation to investigate the role of carbonate-silicate reaction and carbonate composition on the Ca and Mg isotopic signatures in the mantle, and whether these processes can produce detectable light isotopic signals. This model requires several assumptions:

- Initial weight ratio of carbonate and surrounding silicate phase assemblage: **assume a generous upper bound of 1/10 carbonate: pyrolite used in previous calculation for upper mantle conditions (Wang et al., 2014)**
- Amount of carbonate that reacts with the surrounding mantle: **model over range from 0-100% over the subducted pressure range**
- Composition and isotope ratios of the carbonate (Mg/Ca ratio): **model over range from pure MgCO_3 to pure CaCO_3 , isotope ratios per literature (Table 1)**
- Composition and isotope ratios of surrounding mantle: **assume pyrolite composition (Table 1)**
- Equilibrium stable isotope fractionation factors for Mg and Ca isotopes in exchange reactions between carbonates and silicates: **as no measurements for mantle compositions exist, assume values from literature for surface conditions (Table 1) do not change with depth and polymorphism in the carbonates and silicates**

Table 1. Parameters for isotopic mass balance calculations

Parameter	Value	Reference
$\delta^{44/40}\text{Ca}_{\text{pyro}}^i$	0.9 ‰	Kang et al. (2017)
$\delta^{44/40}\text{Ca}_{\text{carb}}^i$	-1.0 ‰	Fantle and Tipper (2014)
$\delta^{26}\text{Mg}_{\text{pyro}}^i$	-0.25 ‰	Teng et al. (2010)
$\delta^{26}\text{Mg}_{\text{carb}}^i$	-4.0 ‰	Wombacher et al. (2011)
CaO abundance in pyrolite	3.17 %	Workman and Hart (2005)
MgO abundance in pyrolite	38.73 %	Workman and Hart (2005)
$\Delta^{44/40}\text{Ca}_{\text{pyro-carb}}$	-0.05 ‰	Amsellem et al. (2020)
$\Delta^{26}\text{Mg}_{\text{pyro-carb}}$	0.06 ‰	Macris et al. (2013)

If no carbonate reaches the lower mantle in subducted slabs, lower-mantle-derived silicates will have heavy $\delta^{44/40}\text{Ca}$ and $\delta^{26}\text{Mg}$, and there would be no lower-mantle-derived carbonate.

If some carbonate reaches the shallow lower mantle in subducted slabs, lower-mantle-derived silicates from this region may have light $\delta^{44/40}\text{Ca}$ due to reaction of persistent metastable CaCO_3 with MgSiO_3 to produce CaSiO_3 (reaction CaC-MgC). The mass balance calculation (Figures below) indicates that a generous upper bound on the masses involved **could locally enrich mantle CaSiO_3 in light $\delta^{44/40}\text{Ca}$** (Fig. 1a). There would be no effect on $\delta^{26}\text{Mg}$ of the silicates in the mantle, the CaC-MgC reaction would make $\delta^{26}\text{Mg}$ in the carbonate relatively heavy (Fig. 1b) and would no longer appear to have a subduction source.

If some carbonate continues to the deeper lower mantle where the reaction MgC-CaC becomes favorable, the much greater abundance of Mg in mantle silicates means that $\delta^{26}\text{Mg}$ of silicates would not be significantly impacted by the breakdown of MgCO_3 (Fig. 2a). However, the $\delta^{44/40}\text{Ca}$ in CaCO_3 produced by this reaction would reflect the heavy mantle source (Fig. 2b), and could be distinguished by CaCO_3 with a surface origin. The exchange reactions potentially overwrite the isotope signals in subducted carbonates with heavy isotopes, and could not significantly affect mantle silicate isotope ratios in the deep lower mantle.

Mg and Ca isotope composition of carbonated pyrolite after the equilibrium isotopic fractionation between carbonate and surrounding pyrolitic mantle depends on the mole ratio of Mg and Ca of the subducted carbonates, but less strongly depends relies on the reaction completion (Fig. 3).

In summary, this first scenario only produces potentially observable heterogeneity in silicate cation isotopes relative to the null case in the CaC-MgC regime, and only in $\delta^{44/40}\text{Ca}$.

- 2) Diamond inclusions formed by redox breakdown of carbonate may trap carbonate or other minerals that can be linked to a deep origin, and be returned to the surface in upwellings.

While our experiments demonstrate that CaCO_3 may form in slabs in the lowermost mantle, CaCO_3 observed in mantle-derived rocks or diamond inclusions would not be a unique signature of the lowermost mantle. However, as explained above, CaCO_3 formed by a cation exchange reaction between MgCO_3 and CaSiO_3 in the lower mantle can be expected to have a different isotope signature relative to subducted CaCO_3 formed at Earth's surface. CaCO_3 formed in the deep lower mantle would contain Ca isotopes that sample the “ambient mantle” source rather than a subducted carbonate source. This could provide a test for shallow vs. ultradeep origin of carbonate inclusions. This is motivation for systematic study of the isotope signatures of diamond inclusions, but available data are insufficient.

We revise the text starting line 179, and add the calculations to the supplementary information, Supplementary Note, Supplementary Fig. 12-13 (corresponding to Fig. 1-3 in this file), Supplementary Table 2 (corresponding to Table 1 in this file).

CaC-to-MgC

Figure 1. Calculated isotopic composition versus reaction rate after the reaction CaC-MgC. n represents the mole fraction of Mg in $(\text{Mg}_n\text{Ca}_{n-1})\text{CO}_3$. Black dashed line represents the average $\delta^{44/40}\text{Ca}$ value in carbonates.

MgC-to-CaC

Figure 2. Calculated isotopic composition versus reaction rate after the reaction MgC-CaC. n represents the mole fraction of Mg in $(\text{Mg}_n\text{Ca}_{n-1})\text{CO}_3$. Black dashed line represents the average $\delta^{26}\text{Mg}$ value in carbonates.

Figure 3. Calculated isotopic composition of carbonated pyrolyte after isotopic fractionation between carbonates and silicates for the reaction (a) CaC-to-MgC and (b) MgC-to-CaC. k represents reaction rate.

I do not think that my comments suggest that the paper is not correct or should not be published. I just think more care should be given to these very high level statements as the data in the paper do not support them.

We appreciate the reviewer's support for publication in Nature Communications and hope you agree that our revisions appropriately develop and clarify the state of knowledge about these potential implications.

Reviewer #2 (Remarks to the Author):

The authors describe novel chemistry of carbonates and silicates at mantle conditions with significant impact for our understanding of geological compositions. The experiments are excellent in their design and appear well-performed, but some improvements in transparency are required by the authors on this front.

Thanks for recognizing the importance and quality of our work.

There is a distinct lack of structural refinement in the article or its supplemental document. Instead, the authors show raw diffraction data alongside tick marks from simulated spectra. While it is appreciable that these systems are mixtures of low-symmetry phases, making structural refinement a challenge, it can also be argued that since these systems are mixtures of low-symmetry phases, a collections of tick marks placed alongside the raw spectrum is not sufficient to show agreement between the model and the observed data.

Thanks for the suggestion. Here we provide two examples of the full-profile XRD fitting, run #1 (Fig. 4) and run #5 (Fig. 5), using Le Bail technique (Le Bail, 2012) implemented in the EXPGUI/GSAS software package (Toby, 2001). The fittings indicate all the identified phases can account for the peaks and intensities of XRD patterns. The reaction products of run#1 contain six phases indicating reaction occurred, and the phase assemblages of run #5 can be fitted with Pv and CaCO_3 indicating no reaction occurred. However, it is noticeable that peaks from different phases are overlapped, which may result in the observed fitting residuals. Therefore, we recognized the peaks and fitted the pattern phase by phase based on its equation of state and crystal structure, using the program PDIndexer (Seto et al., 2010).

We add these representative full profile fittings to the supplementary information, Supplementary Fig. 5 (corresponding to Fig.4-5 in this file).

Figure 4. Full-profile fitting for XRD of run #1.

Figure 5. Full-profile fitting for XRD of run #5.

In addition, it should be common practice to show microscope images of loaded samples at pressure in diamond cell experiments. It is important that the integrity of the experiment can be subject to scrutiny by the referees and eventual readers, and the sample loading is central to the success and quality of any diamond cell experiment. These authors are at the top of the field of static high pressure, and I hope that they agree, and will provide these in their supplement and any future publications.

Thanks for the suggestion. Here we use run #9 as an example to show more details about sample loading, images of sample at high pressure, and temperature measurements. We loaded the Fe-bearing sample on top of the thermal insulation layer on the piston side of DAC (75/300 beveled anvil), then we loaded another insulation layer on the cylinder side of DAC together with Re gasket before we close and compress the DAC to the target pressure (Fig. 6a). The size of the sample is smaller than the two thermal insulation layers in order to be well insulated from diamond and gasket. Under the target pressure before heating, the sample showed gray-to-black color within the transparent thermal insulation layers, and the crosshair in the red square indicates the position that we heated (Fig. 6b). The sample was heated using a double-sided ytterbium fiber laser heating system, and the round and stable heating spot indicates a good heating quality (Fig. 6c).

Temperatures of the heated samples were determined by fitting the measured thermal radiation spectra using the Planck radiation function under the graybody approximation, and a typical example of profile fittings for upstream and downstream temperature measurements are shown in Fig. 7.

We add these representative images and fittings to the supplementary information, Supplementary Fig. 2 (corresponding to Fig.6 in this file) and Supplementary Fig. 11 (corresponding to Fig.7 in this file).

Figure 6. Microscope images of loaded sample for run #9. (a) Samples are loaded at ambient conditions on the piston side before closing the cell. (b) Samples are compressed to target pressure before heating, and the dashed circle indicates the dark Fe-bearing sample. (c) The heating spot on the loaded sample during laser heating.

Figure 7. Typical temperature measurements and fitting profiles of upstream and downstream for run #9.

The work is scientifically sound from beginning to end, and has implications which are of broad enough interest for the readership of Nature Communications. The authors demonstrate suitable attention to the literature and provide proper discussion of their results in view of some of the most recent results in carbonate chemistry at these conditions. Overall, the work demonstrates the state-of-the-art for LH-DAC experiments, but not the cutting-edge. Based on the experimental findings, the work warrants publication in Nature Communications, I would only ask that – especially for a broad readership journal – the authors provide more detailed illustration and documentation of their experimental procedures for the non-specialist.

Some notes on that front:

- The aforementioned structural fitting of XRD data

- Images of sample loadings
- Representative Planck fitting for temperature derivation

We appreciate the reviewer's support for publication in Nature Communications and have included these new figures.

References

- Amsellem, E., Moynier, F., Bertrand, H., Bouyon, A., Mata, J., Tappe, S., & Day, J. M. D. (2020). Calcium isotopic evidence for the mantle sources of carbonatites. *Sci Adv*, 6(23), eaba3269. <http://10.1126/sciadv.aba3269>
- Fantle, M. S., & Tipper, E. T. (2014). Calcium isotopes in the global biogeochemical Ca cycle: implications for development of a Ca isotope proxy. *Earth-Science Reviews*, 129, 148-177.
- Kang, J.-T., Ionov, D. A., Liu, F., Zhang, C.-L., Golovin, A. V., Qin, L.-P., et al. (2017). Calcium isotopic fractionation in mantle peridotites by melting and metasomatism and Ca isotope composition of the Bulk Silicate Earth. *Earth and Planetary Science Letters*, 474, 128-137.
- Le Bail, A. (2012). Whole powder pattern decomposition methods and applications: A retrospection. *Powder Diffraction*, 20(4), 316-326. <http://10.1154/1.2135315>
- Macris, C. A., Young, E. D., & Manning, C. E. (2013). Experimental determination of equilibrium magnesium isotope fractionation between spinel, forsterite, and magnesite from 600 to 800 C. *Geochimica et Cosmochimica Acta*, 118, 18-32.
- Seto, Y., Nishio-Hamane, D., Nagai, T., & Sata, N. (2010). Development of a software suite on X-ray diffraction experiments. *The Review of High Pressure Science and Technology*, 20(3), 269-276. <http://10.4131/jshpreview.20.269>
- Teng, F.-Z., Li, W.-Y., Ke, S., Marty, B., Dauphas, N., Huang, S., et al. (2010). Magnesium isotopic composition of the Earth and chondrites. *Geochimica et Cosmochimica Acta*, 74(14), 4150-4166.
- Toby, B. H. (2001). EXPGUI, a graphical user interface for GSAS. *Journal of Applied Crystallography*, 34(2), 210-213. <http://10.1107/S0021889801002242>
- Wang, S. J., Teng, F. Z., & Li, S. G. (2014). Tracing carbonate-silicate interaction during subduction using magnesium and oxygen isotopes. *Nat Commun*, 5, 5328. Article. <http://10.1038/ncomms6328>
- Wombacher, F., Eisenhauer, A., Böhm, F., Gussone, N., Regenber, M., Dullo, W.-C., & Rüggeberg, A. (2011). Magnesium stable isotope fractionation in marine biogenic calcite and aragonite. *Geochimica et Cosmochimica Acta*, 75(19), 5797-5818.
- Workman, R. K., & Hart, S. R. (2005). Major and trace element composition of the depleted MORB mantle (DMM). *Earth and Planetary Science Letters*, 231(1-2), 53-72. <http://10.1016/j.epsl.2004.12.005>

REVIEWERS' COMMENTS

Reviewer #1 (Remarks to the Author):

I think the authors did a great job of responding to my comments. Paper looks great.

Reviewer #2 (Remarks to the Author):

The authors have addressed each of my previous comments in a satisfactory manner. With the addition of full-profile structural refinement information, microscope images of the sample environment, and information on laser heating techniques, this article now has an acceptable level of transparency.

Comments from the other referee regarding the accuracy of geological language have been adequately handled in this revised manuscript.

I would be happy to see this article published in Nature Communications in its current form.